# Effect of Solution Treatment Time on Microstructure Evolution and Properties of Mg-3Y-4Nd-2Al Alloy

**DOI:** 10.3390/ma16062512

**Published:** 2023-03-22

**Authors:** Lili Zhao, Sicong Zhao, Yicheng Feng, Lei Wang, Rui Fan, Tao Ma, Liping Wang

**Affiliations:** 1School of Material Science and Chemical Engineering, Harbin University of Science and Technology, Harbin 150000, China; 2Key Laboratory of Advanced Manufacturing and Intelligent Technology (Ministry of Education), Harbin University of Science and Technology, Harbin 150000, China; 3College of Light Industry, Harbin University of Commerce, Harbin 150028, China

**Keywords:** Mg-3Y-4Nd-2Al, solution treatment, phase evolution, grain size, plasticity

## Abstract

In order to explore the microstructure evolution of an Mg-RE alloy refined by Al during solution treatment, an Mg-3Y-4Nd-2Al alloy was treated at 545 °C for different time periods. Phase evolution of the alloy was investigated. After solution treatment, the Mg-RE eutectic phase in the Mg-3Y-4Nd-2Al alloy dissolves, the granular Al_2_RE phase does not change, the acicular Al_11_RE_3_ phase breaks into the short rod-like Al_2_RE phase, and the lamellar Al_2_RE phase precipitates in the grains. With the extension of solution time, the precipitated phase of the lamellar Al_2_RE increased at first and then decreased, and its orientation relationship with the matrix is ${<112>}_{{\mathrm{Al}}_{2}\mathrm{RE}}{//<2\bar{1}\bar{1}0>}_{\mathrm{Mg}}$ and ${\left\{111\right\}}_{{\mathrm{Al}}_{2}\mathrm{RE}}{//\{0002\}}_{\mathrm{Mg}}$. The undissolved granular Al_2_RE phase can improve the thermal stability of the alloy grain by pinning the grain boundary, and the grain size did not change after solution treatment. Solution treatment significantly improved the plasticity of the alloy. After 48 h of solution treatment, the elongation increased to 17.5% from 8.5% in the as-cast state.

## 1. Introduction

Cast Mg-Y-Nd alloy has been widely used in industrial fields because of its excellent mechanical properties at room temperature and high temperatures [1,2,3,4]. The cast Mg-RE alloys without refinement have very well-developed and coarse grains, making their mechanical properties low while promoting the formation of defects such as hot cracking and porosity. Therefore, grain refinement is necessary for the cast Mg-RE alloy [1,4]. Generally, Zr is used as a grain refiner of the Mg-Y-Nd alloy [5,6]. However, Zr refinement of the Mg-Y-Nd alloy has the disadvantage of low efficiency and high cost. In recent years, researchers have found that the addition of Al instead of Zr as a refiner is an effective strategy for the Mg-Y-Nd alloy [7,8].

Most Mg-RE alloys need to be strengthened by heat treatment. Solution treatment, as an indispensable pretreatment process, provides the microstructure basis for aging treatment [9,10,11]. Therefore, controlling the microstructure transformation in the solution process is one of the key factors to improve the mechanical properties. For the Zr-refined Mg-RE alloy, the second phase in the as-cast state is mainly the Mg-RE eutectic phase, and the Mg-RE phase completely dissolves into the matrix after solution treatment [12]. For the Al-refined Mg-RE alloy, due to the lower enthalpy of formation between Al and RE elements, various second phases can be formed, such as Al_2_RE, Al_11_RE_3_, LPSO, et al. [13,14,15,16]. Ding et al. [17] showed that the Al_2_RE phase in the Mg-RE-Al alloy is stable at high temperatures, which cannot be dissolved by homogenization and solution treatment. Su et al. [18] revealed that the Al_11_RE_3_ phase in the Mg-RE-Al alloy at high temperatures is unstable and can be decomposed to form the Al_2_RE phase. Li et al. [19] found that the Mg-10Y-1Al alloy has excellent thermal stability after solution treatment. The reason for this stability is that the long-period stacking-ordered (LPSO) phase precipitated along the grain boundary can effectively pin the grain boundary. In addition, Peng et al. [20] found that after solution treatment, in addition to the micro-sized Al_2_Y particles in or along the grain boundary, nano-sized particles would also precipitate in or along the grain boundary of the Mg-1Al-6.2Y alloy, and confirmed that the precipitated phase was Al_2_Y.

In summary, different from the traditional Zr-refined Mg-RE alloy, the microstructure change in Al-refined Mg-RE alloy during heat treatment is more complex. At present, the research on the microstructure evolution of new Al-refined Mg-RE alloy during solution treatment is very limited. In view of the importance of solution treatment of the Mg-RE-Al alloy, the microstructure and evolution of the Mg-3Y-4Nd-2Al alloy during solution treatment are deeply analyzed in this paper. In addition, the effect of solution time on the thermal stability and tensile properties of alloy grains is also discussed, which provides a theoretical basis for the development of the heat treatment process of the Mg-RE-Al alloy.

## 2. Experimental Method

The experimental alloy selected in this paper is the Mg-3Y-4Nd-2Al alloy. The experimental alloy is made of pure Mg, pure Al, Mg-30 wt.% Nd, and Mg-30 wt.% Y master alloy. All raw materials were smelted in a resistance furnace with a steel crucible. The mixture of SF_6_ (1 vol.%) and CO_2_ (99 vol.%) was used to protect the melting process, and the melting temperature was 750 °C. The melt was stirred, rested, slagged, and then poured into a metal mold preheated at 200 °C (cavity size: 100 mm × 10 mm × 60 mm). A box-type resistance furnace (HCY-03F, Songjiang Co., Ltd., Harbin, China) was used, the temperature control precision was less than ±2 °C, and the solution treatment temperature was 545 °C based on the result of the differential scanning calorimeter (DSC) (STA 449F3, Netzsch, Bavaria, German). The DSC samples were heated from room temperature to 700 °C at the rate of 10 °C/min. The solution treatment time was from 2 h to 48 h, and the water medium was used for cooling.

The samples were etched by using picric acid caustic (10 g picric acid +8 mL glacial acetic acid +20 mL deionized water +70 mL ethanol). The microstructure was observed by optical microscope (OM, XD30M, Beijing Instant Hengye Technology Co., Ltd., Beijing, China), and the average grain size was measured by the linear intercept method. The phase crystal structure of the alloy was analyzed by X-ray diffractometer (XRD)(X’Pert PRO, PANalytical B.V., Almelo, The Netherlands). The experimental voltage was set at 40 kV, Cu target was used, the scanning range was 10°~90°, and the scanning rate was 8°/min.

Scanning electron microscopy (SEM, Apreo C, Thermo Fisher Scientific Inc., St. Bend, OR, USA) was used to observe the morphology and distribution of the second phase. Transmission electron microscopy (TEM, TALOS 200FX G2, Thermo Fisher Waltham, Waltham, MA, USA) was used to observe the microstructure of the alloy and analyze the structure and element distribution of the second phase in the alloy. The TEM specimens were ground to 40~50 μm in thickness and then prepared by ion milling operating at 20 μA and 3°~9° milling angle.

The universal testing machine (MTS E44.304, MTS Systems Co., Eden Prairie, MN, USA) was used to carry out the tensile test at room temperature at a tensile rate of 1 mm/min. The gauge dimension of the tensile specimens is 15 mm × 3 mm × 2 mm. In order to ensure the test accuracy, five tensile samples were measured, and the average value was taken as the final test result.

## 3. Results and Analysis

### 3.1. Microstructure of As-Cast Alloy

Figure 1a–c are SEM, XRD, and EDS energy spectra of the as-cast Mg-3Y-4Nd-2Al alloy, respectively. Combined with various characterization methods, it is known that the as-cast Mg-3Y-4Nd-2Al alloy mainly contains three kinds of second phases, namely, the granular phase in the grain (Al_2_RE phase), the acicular second phase (Al_11_RE_3_ phase) near the grain boundary, and the eutectic phase near the grain boundary (Mg-RE intermetallic phase: Mg_12_RE, β_1_-Mg_14_Nd_2_Y). The Al_2_RE phase is preferentially precipitated during solidification and can be used as the nucleating particles to refine grains [21].

### 3.2. DSC of As-Cast Alloy

In order to determine the dissolution temperature of the phase in the alloy, the as-cast Mg-3Y-4Nd-2Al alloy was analyzed by DSC, and the results are shown in Figure 2. The DSC curve of the Mg-3Y-4Nd-2Al alloy has one large and one small endothermic peak. The larger endothermic peak is at 640.5 °C, corresponding to the dissolution temperature of the Mg matrix. The smaller endothermic peak is at 547.6 °C, corresponding to the dissolution temperature of the second phase. To avoid overheating, the solid solution temperature was selected near the solution temperature of the second phase. Thus, the solid solution temperature was 545 °C.

### 3.3. Phase Evolution of Solution-Treated Alloy

Figure 3 shows the XRD patterns of the Mg-3Y-4Nd-2Al alloy after solution treatment at 545 °C for different time periods. The diffraction peaks of the α-Mg, Al_2_RE, Mg_12_RE, and Al_11_RE_3_ phases were mainly contained in the alloy after 2 h of solution treatment. When the solution time was longer than 2 h, only α-Mg and Al_2_RE phase diffraction peaks were contained in the alloy, indicating that only the Al_2_RE phase existed in the alloy after solution treatment. The Al_2_RE phase has a high melting point and does not dissolve during solution treatment. Similarly, there is an undissolved second phase in the Al-refined Mg-RE alloy studied by Ding [17] after solution treatment.

In order to further characterize the microstructure evolution of the Mg-3Y-4Nd-2Al alloy during solution treatment, the solution-treated alloy was analyzed by SEM, and the results are shown in Figure 4. After solution treatment for 2 h, the alloy contained a small amount of undissolved Mg-RE eutectic phase, as shown by the blue arrow in Figure 4a. The morphology of the granular Al_2_RE phase (“A” phase) and the acicular Al_11_RE_3_ phase (“B” phase) was almost unchanged, while a large number of the fine lamellar phase (“C” phase) was precipitated within the grains. As shown in Figure 4b–f, with the extension of solution time, the Mg-RE eutectic phase was almost completely dissolved into the Mg matrix, while the granular phase remained unchanged, and the acicular phase gradually evolved into a short rod-like morphology and gradually spheroidized at the grain boundary. Only the second phase with a partially short rod-like morphology can be observed at the grain boundary. When the solution time is extended to 4 h (Figure 4b), the lamellar second phase precipitated inside the grain boundary increases more densely, while when the solution time reaches 8~16 h (Figure 4c,d), the number of the lamellar second phase decreases and the size increases. When the solution time is extended to 24 h to 48 h (Figure 4e,f), the content of the lamellar phase is very little. Combined with the results of SEM and XRD analyses, the short rod-like phase and precipitated lamellar phase after solution treatment may be the Al_2_RE phase.

In order to further determine the structure of each phase in the solution-treated alloy, TEM analysis was carried out. Figure 5 shows the TEM image and SAED patterns of granular phases in the solution-treated alloy. After the solution treatment, the morphology of the granular phase does not change obviously, and the granular phase has a face-centered cubic structure, and the lattice constant is about 0.78 nm, which is determined to be the Al_2_RE phase.

Figure 6 shows the HAADF-STEM morphology, SEAD patterns, and EDS mapping of the acicular phase before and after fracture with different solution times. As shown in Figure 6a, when the solution time is 2 h, the morphology of the acicular phase is basically the same as that of the as-cast alloy, but some of the acicular phase has broken and transformed into a short rod-like phase. From Figure 6b, it can be seen that both the acicular phase and the short rod-like phase mainly contain Y, Nd, and Al elements. SAED analysis shows that the acicular phase is the Al_11_RE_3_ phase, while the short rod-like phase is the Al_2_RE phase, indicating that the Al_11_RE_3_ phase is metastable in Mg-Y-Nd-Al alloys. The fractured acicular phase with a solution time of 4 h is shown in Figure 6c,d. Combined with the mapping and SEAD results, it can be seen that the fractured acicular phase is still the Al_2_RE phase, indicating that when the solution time exceeds 2 h, all the acicular phases fracture and transform into the Al_2_RE phase, which is consistent with the XRD results.

The existence of precipitated phase in the Al-refined Mg-RE alloy during solid solution is an interesting phenomenon [13,14,15,16,17]; however, the precipitated phase was not characterized in detail in previous studies. Therefore, the lamellar precipitated phase was analyzed and studied in detail by TEM technology in this paper. Figure 7 shows the HAADF-STEM morphology, EDS mapping, HRTEM images, and SAED patterns of the lamellar phase of the alloy with different solution times. It can be seen that the lamellar phase can be observed in the matrix under different solution times as shown in Figure 7a,e,i and the lamellar phase mainly contains Y, Nd, and Al elements as shown in Figure 7b,f,j. HRTEM images showed that there was a clear interface between the lamellar precipitated phase and the α-Mg matrix as shown in Figure 7c,g,k. It can be confirmed from SAED patterns that the lamellar phase is the Al_2_RE phase, and the orientation relationship between the Al_2_RE precipitated phase and the matrix is: ${\left[\bar{1}12\right]}_{{\mathrm{Al}}_{2}\mathrm{RE}}//[11\bar{2}{0]}_{\mathrm{Mg}}$, ${\;(\bar{1}1\bar{1})}_{{\mathrm{Al}}_{2}\mathrm{RE}}{//\{0001\}}_{\mathrm{Mg}}$ as shown in Figure 7d,h,l.

Because of the high solubility of RE in the α-Mg at high temperatures [22], the Mg-RE eutectic phase in the Mg-3Y-4Nd-Al alloy dissolved and the RE element diffused into the α-Mg matrix to form a solid solution after the solution treatment. The previous results show that the Al-RE phase has the lowest enthalpy of formation in Mg-RE-Al alloys, while the Al_2_RE phase has the lowest enthalpy of formation [23,24] in the Al-RE phase; therefore, the Al_2_RE phase is the most stable. Because the formation enthalpy of the Al_11_Nd_3_ phase is higher than that of the Al_2_RE phase, it decomposed and transformed into the Al_2_RE phase during the solution treatment. Zhang et al. [25,26] showed that when the temperature is higher than 150 °C, the Al_11_RE_3_ phase in the alloy decomposes according to the following reactions:Al_11_RE_3_→3Al_2_RE + 5Al(1)

The formation of the lamellar Al_2_RE phase during the solution treatment is related to the fracture of the acicular Al_11_RE_3_ phase and the dissolution of the Mg-RE phase. The Al element released by the decomposition of the acicular Al_11_RE_3_ phase combines with the RE element released by the dissolution of the Mg-RE eutectic phase to precipitate the lamellar phase in the crystal. When the solution time is 2~4 h, the number of lamellar precipitates is large, and their size is fine. With the extension of the solution time, the number of intragranular lamellar phases decreases, and the size increases.

According to the principle of solid phase transformation, the new phase always precipitates along the specific crystal plane of the matrix, that is, there is a habit plane to ensure the minimum resistance to phase transformation [27]. According to the calculation of the E2EM model [28], the mismatch between the close-packed plane {111}_Al2RE_ of the Al_2_RE and {0002}_Mg_ of the α-Mg is the minimum, and the mismatch between the close-packed direction <112>_Al2RE_ and ${<11\bar{2}0>}_{\mathrm{Mg}}$ is the minimum. Therefore, when the precipitated Al_2_RE phase and the α-Mg matrix have the orientation relationship ${\;<112>}_{{\mathrm{Al}}_{2}\mathrm{Nd}}{//<11\bar{2}0>}_{\mathrm{Mg}}$, ${\left\{111\right\}}_{{\mathrm{Al}}_{2}\mathrm{Nd}}{//\{0002\}}_{\mathrm{Mg}}$, the strain energy is the lowest, and the precipitation is the easiest. Consequently, the habit plane is (0002)_Mg_ and the characterization in Figure 7d,h,l confirms that the precipitated Al_2_RE satisfies the orientation relationship.

### 3.4. Grain Thermal Stability

Al refinement of the Mg-RE alloy usually has good grain thermal stability. Figure 8 shows the change in grain size of the as-cast alloy with different solution times. The average grain size of the as-cast alloy is 49 ± 4 μm. With the extension of solution time, the grain size does not change obviously. After holding for 48 h, the average grain size is 51 ± 5 μm. It can be seen that, as reported in the literature [19], the Mg-3Y-4Nd-2Al alloy also shows excellent grain thermal stability. It can be seen that the Mg-3Y-4Nd-2Al alloy shows excellent grain thermal stability during solution treatment at 545 °C. In general, the undissolved second phase at the grain boundary could hinder the growth of grains in the process of heat treatment. As shown in Figure 9a, in the solution-treated alloy, the granular, short rod-like and lamellar phases can be observed at the grain boundary. Through the EDS mapping shown in Figure 9b, these phases are all the Al_2_RE phase. These high-temperature stable Al_2_RE phases can effectively pin the grain boundary, hinder the movement of the grain boundary, and improve the thermal stability of grains.

### 3.5. Mechanical Properties of Solution-Treated Alloys

Figure 10 shows the tensile properties of the Mg-3Y-4Nd-2Al alloy after the solution treatment at 545 °C for different time periods. The results show that the yield strength and tensile strength of the as-cast alloy are 125 MPa and 205 MPa, respectively, and the elongation is 8.5%. With the extension of the solution time, the yield strength and tensile strength change little and the elongation increases greatly. After a solution treatment of 48 h, the elongation is 17.5%, which is 106% higher than that of the as-cast alloy.

The strengthening mechanism of the as-cast alloy is mainly fine-grain strengthening and second-phase strengthening. The contribution of fine grain strengthening remains unchanged because the grain size of the alloy does not change obviously after solution treatment. After solution treatment, the second phase strengthening decreases due to the dissolution of the Mg-RE eutectic phase, and the lamellar phase precipitated along the base plane has a larger size and has little effect on the strength. According to the EDS results provided in the Appendix A, the total content of the RE elements in the matrix of the as-cast alloy is 0.37 at.%. After holding at 545 °C for 48 h, the total content of the RE elements in the matrix is 0.62 at.%. Therefore, the solid solution strengthening effect is enhanced due to the increase in solute elements in the matrix. Because the strength of the alloy changes little before and after solution treatment, it shows that the decrease in the second phase strengthening of the alloy after solution treatment is close to the increase in solid solution strengthening.

Figure 11 shows the tensile fracture morphology of the Mg-3Y-4Nd-2Al alloy after solution treatment at 545 °C for different time periods. As shown in Figure 11a, the fracture of the as-cast alloy is mainly composed of cleavage planes, tearing edges, and granular protrusions. The fracture mode is a mixture of intergranular brittle fracture and transgranular fracture. The fracture morphology changed significantly after solution treatment, and the granular protrusion decreased significantly due to the dissolution of the eutectic phase. The fracture surface was mainly composed of cleavage planes, tearing edges, and a small amount of dimples. The fracture mode is a ductile transgranular fracture as shown in Figure 11b–d. With the extension of the solution time, the number of tear edges and dimples gradually increased, which indicated that the plasticity of the alloy increased gradually. In addition, with the extension of the solution time, the number of tear edges and dimples gradually increased, which indicated that the plasticity of the alloy gradually increased.

In the as-cast alloy, because there is a large number of Mg-RE eutectic phases at the grain boundaries, which are large-size brittle intermetallic compounds [29,30], the stress concentration easily occurs along the Mg-RE eutectic phases under the tensile loading, which leads to crack initiation and propagation [31,32,33,34]. As shown in Figure 12a, the cracks in the as-cast alloy mainly originated from the Mg-RE eutectic phases at the grain boundary. Due to the large number of Mg-RE eutectic phases in the as-cast alloy, more cracks were formed, showing insufficient plasticity. After a solution treatment of 2 h, compared with the as-cast alloy, most of the eutectic phase dissolved, the crack source decreased, and the plasticity of the alloy increased. As shown in Figure 12b, there are still a small amount of undissolved eutectic cracks. With the further extension of the solution time, all the eutectic phases dissolved, which greatly reduced the stress concentration at the grain boundary. However, the lamellar phase precipitated on the base plane and its sharp corner also easily form stress concentrations. Thus, when the solution time is 4 h, the plasticity of the alloy is not significantly improved. After 16 h of the solution treatment, the lamellar phase decreased, and the plasticity further increased, but the existence of cracks can still be observed near the larger lamellar phase, as shown in Figure 12c. With the further extension of the solution time to 48 h, the lamellar phase decreased obviously, and the cracks were mainly caused by the granular phase in the grain, as shown in Figure 12d. Because the morphology of the granular phase is approximately circular, the stress concentration is small, and the plastic deformation is larger during fracture. The plasticity of the alloy is further improved. 

## 4. Conclusions

After solution treatment of the Mg-3Y-4Nd-2Al alloy, the Mg-RE eutectic phase dissolves and the granular Al_2_RE phase does not change obviously. With the extension of solution time, the acicular Al_11_RE_3_ phase fractures and evolves into a short rod-like phase. The Al_2_RE lamellar phase precipitates in the grain during solution treatment. With the extension of solution time, the precipitated phase first increases and then decreases. The orientation relationship between the lamellar Al_2_RE phase and the α-Mg is ${<112>}_{{\mathrm{Al}}_{2}\mathrm{RE}}{//<2\bar{1}\bar{1}0>}_{\mathrm{Mg}}$, ${\left\{111\right\}}_{{\mathrm{Al}}_{2}\mathrm{RE}}{//\{0002\}}_{\mathrm{Mg}}$.The undissolved and precipitated Al_2_RE phase during the solution treatment causes the Mg-3Y-4Nd-2Al alloy to have excellent grain thermal stability, and the grain size does not change after the solution treatment of 48 h at 545 °C.With the extension of solution time, the strength of the alloy changed little and the plasticity significantly increased. After the solution treatment of 48 h, the elongation of the alloy increased by 106% compared with the as-cast alloy. The fracture of as-cast alloy is a mixed mode of intergranular fracture and transgranular fracture, the cracks are concentrated at the eutectic phase, and plasticity is insufficient. After the solution treatment, the fracture mode changes to a transgranular fracture, and the cracks are mainly concentrated in the undissolved granular phase. A small amount of cracks occurred at the precipitated lamellar phase.

## Figures and Tables

**Figure 1 materials-16-02512-f001:**
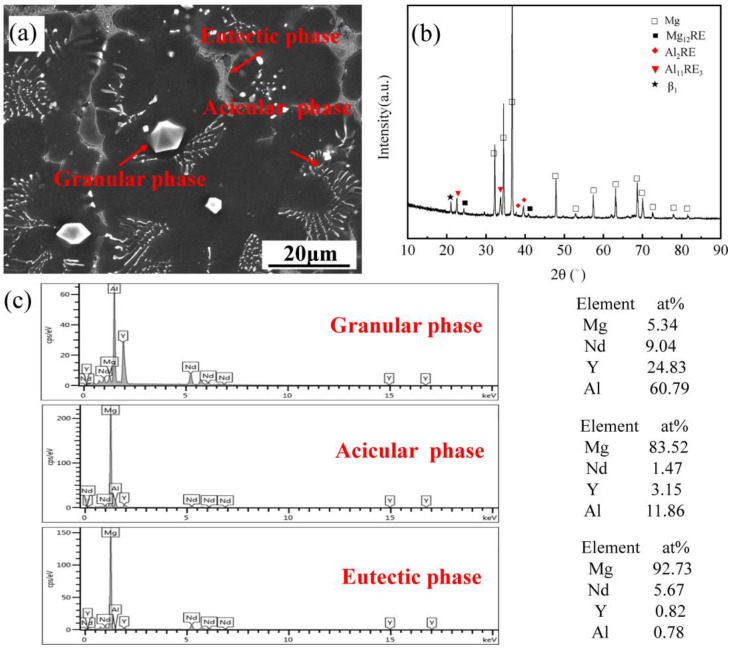
As-cast Mg-3Y-4Nd-2Al alloy: (**a**) SEM, (**b**) XRD pattern, and (**c**) EDS mapping.

**Figure 2 materials-16-02512-f002:**
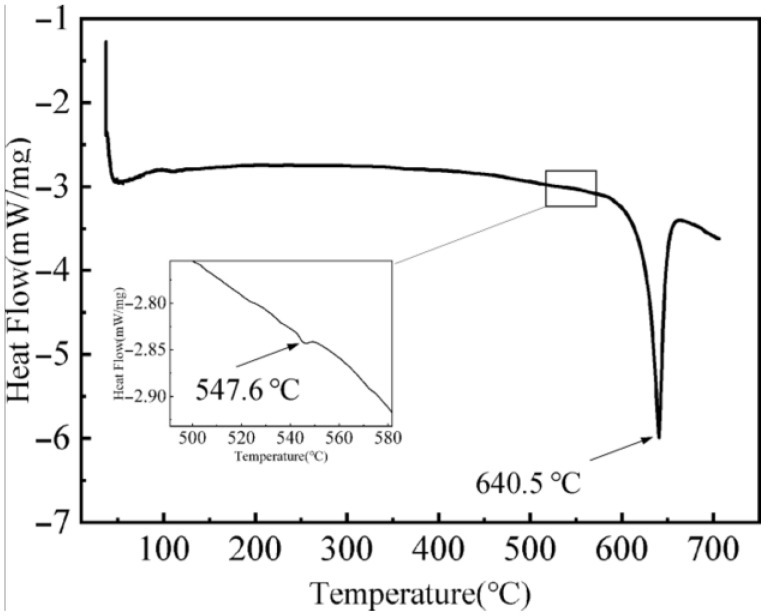
DSC curve of as-cast Mg-3Y-4Nd-2Al alloy.

**Figure 3 materials-16-02512-f003:**
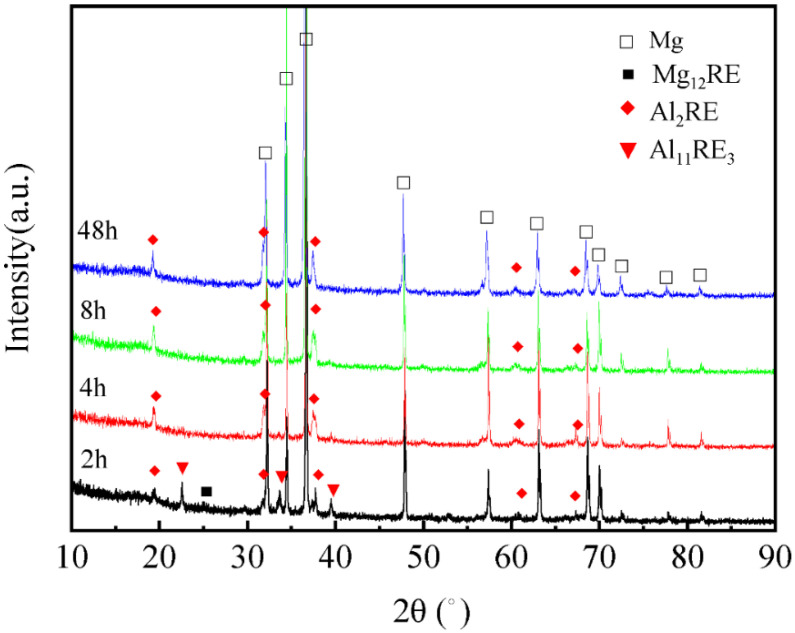
XRD patterns of the Mg-3Y-4Nd-2Al alloy after solution treatment at 545 °C for different time periods.

**Figure 4 materials-16-02512-f004:**
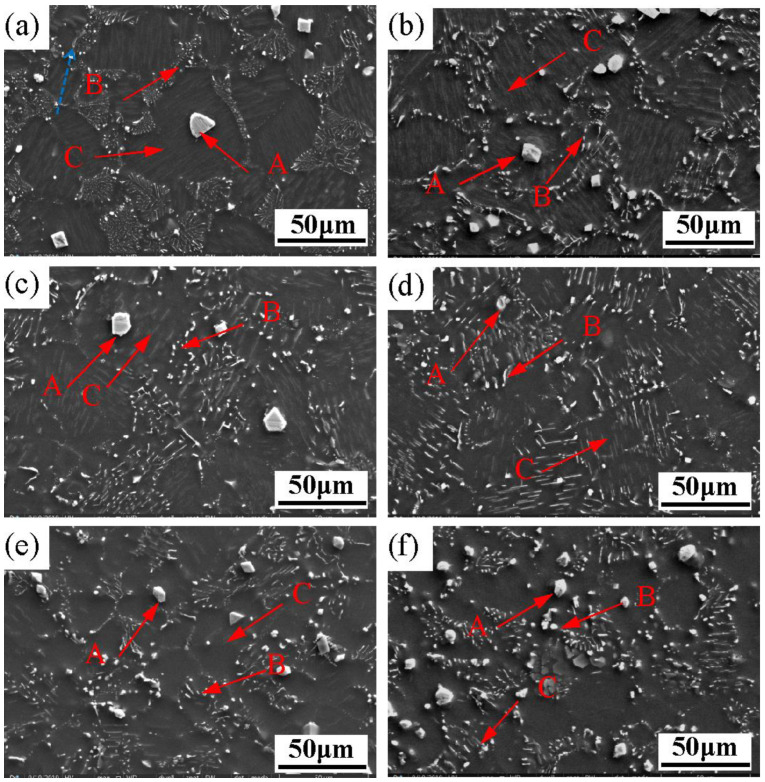
SEM images of the Mg-3Y-4Nd-2Al alloy after solution at 545 °C for different time periods: (**a**) 2 h, (**b**) 4 h; (**c**) 8 h, (**d**) 16 h, (**e**) 24 h, and (**f**) 48 h.

**Figure 5 materials-16-02512-f005:**
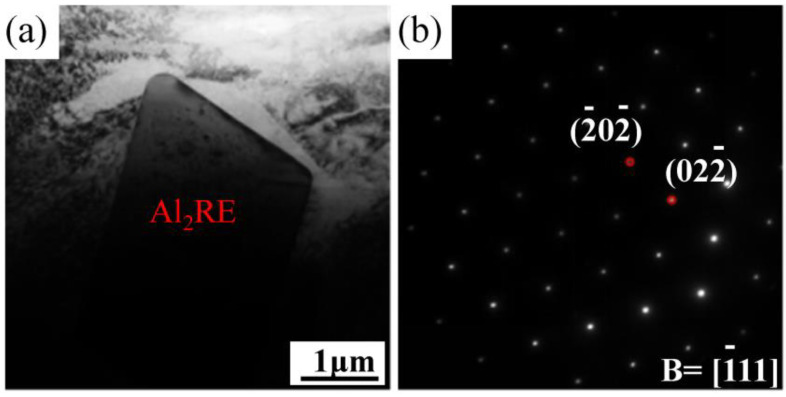
TEM image and SAED patterns of granular Al_2_RE phase in the Mg-3Y-4Nd-2Al alloy after solution treatment at 545 °C: (**a**) TEM image and (**b**) SAED patterns.

**Figure 6 materials-16-02512-f006:**
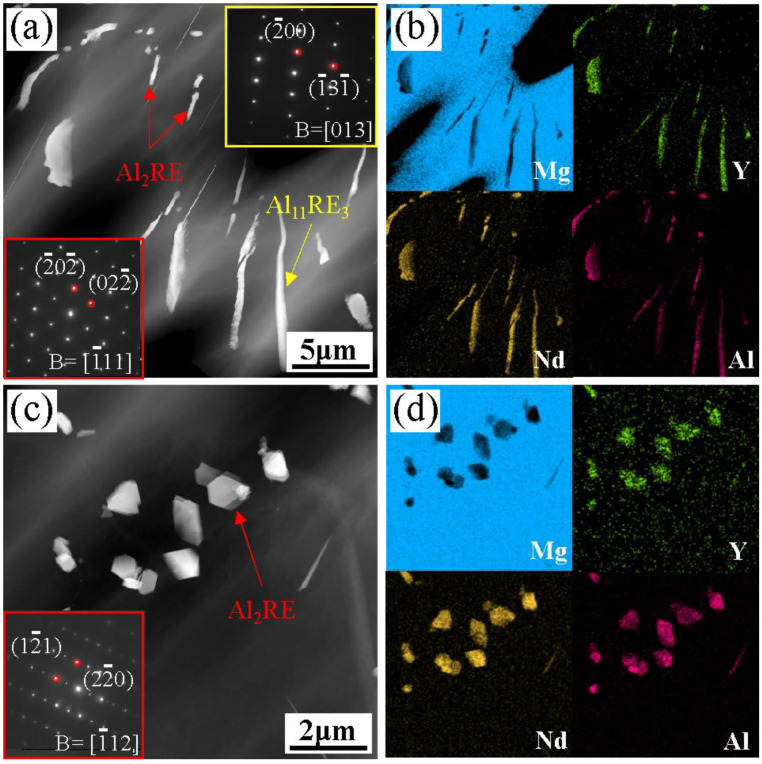
HAADF-STEM morphology, SEAD patterns, and EDS mapping of acicular phase and short rod-like phase in the Mg-3Y-4Nd-2Al alloy with different solution times: (**a**,**b**) 2 h; (**c**,**d**) 4 h.

**Figure 7 materials-16-02512-f007:**
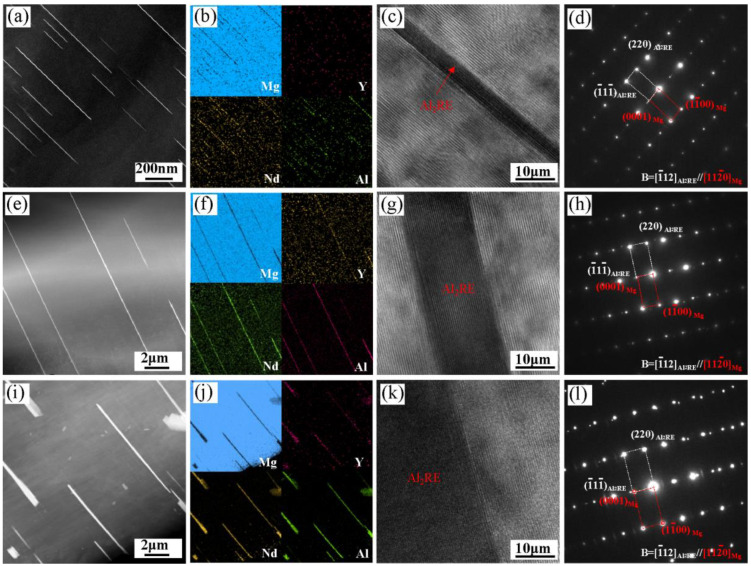
HAADF-STEM morphology, EDS mapping, HRTEM images, and SAED patterns of lamellar phase in the Mg-3Y-4Nd-2Al alloy with different solution times: (**a**–**d**) soluted for 2 h; (**e**–**h**) soluted for 8 h; (**i**–**l**) soluted for 48 h.

**Figure 8 materials-16-02512-f008:**
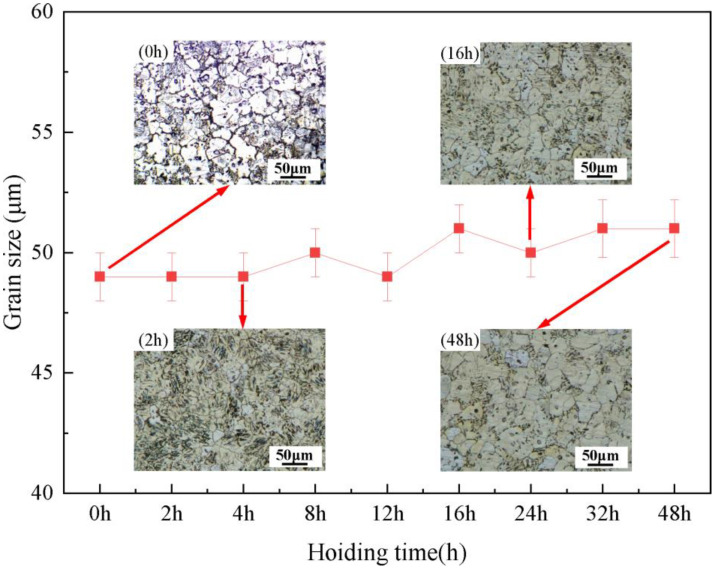
Changes in grain size of the Mg-3Y-4Nd-2Al alloy with different solution times.

**Figure 9 materials-16-02512-f009:**
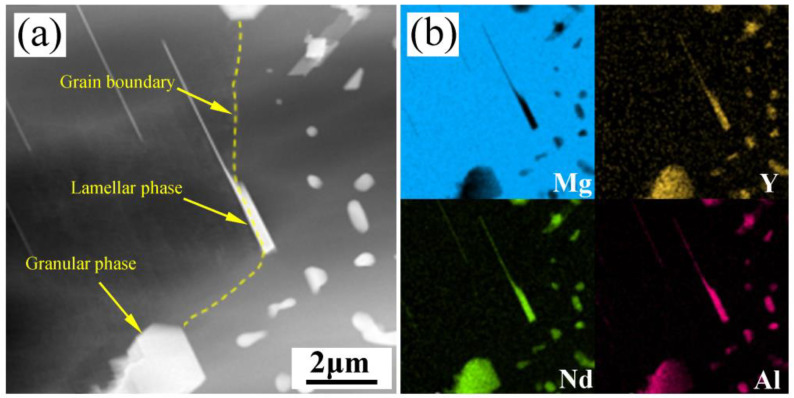
HAADF-STEM morphology and EDS mapping of Al_2_RE phase at the grain boundary of the Mg-3Y-4Nd-2Al alloy after solution treatment for 2 h: (**a**) HAADF-STEM morphology, (**b**) EDS mapping.

**Figure 10 materials-16-02512-f010:**
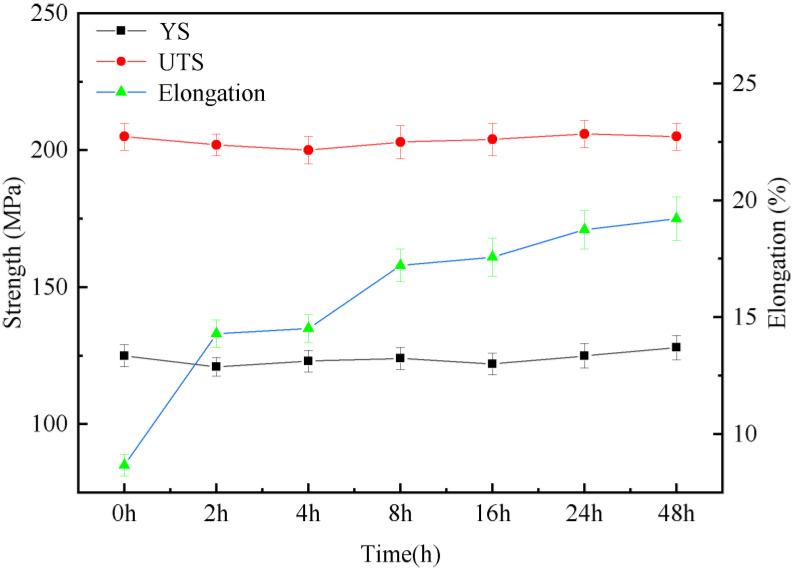
Tensile properties of the Mg-3Y-4Nd-2Al alloy after solution treatment at 545 °C for different time periods.

**Figure 11 materials-16-02512-f011:**
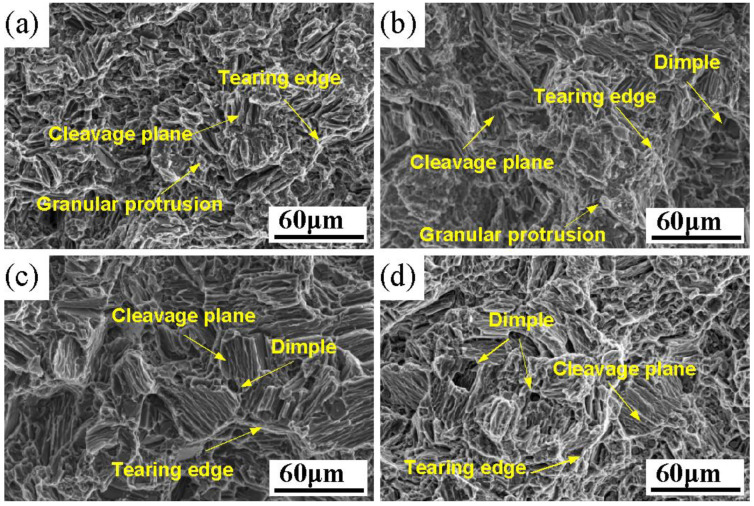
Tensile fracture morphology of the Mg-3Y-4Nd-2Al alloy after solution treatment for different time periods: (**a**) 0 h, (**b**) 2 h, (**c**) 16 h, and (**d**) 48 h.

**Figure 12 materials-16-02512-f012:**
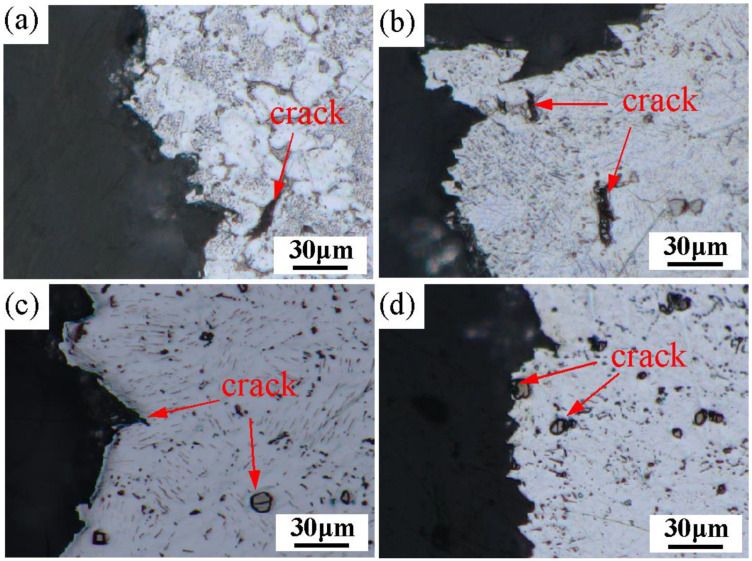
Metallography of side fracture of the Mg-3Y-4Nd-2Al alloy after solution treatment for different time periods:(**a**) 0 h; (**b**) 2 h; (**c**) 16 h; (**d**) 48 h.

## Data Availability

The data presented in this study are available on request from the corresponding author.

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
