# Peer review of "Effect of Solution Treatment Time on Microstructure Evolution and Properties of Mg-3Y-4Nd-2Al Alloy"

_materials, 2023, doi:10.3390/ma16062512_

Round 1
Reviewer 1 Report
The authors investigated the effect of solution treatment on the microstructures and mechanical properties of Mg-3Y-4Nd-2Al alloy. The microstructural characterization is excellent. However, I was unable to grasp the work's purpose. In the soultionized conditions, there are undissolved precipitates. Why did the authors select 545 degrees Celsius as the solutionizing temperature? Why was no aging treatment performed?
Why has strength remained unchanged while ductility has improved?
Author Response
Dear Editor and reviewers:
Thanks a lot for your precious and constructive comments for improving the manuscript entitled "Effect of Solution Treatment Time on Microstructure Evolution and Properties of Mg-3Y-4Nd-2Al Alloy" (Manuscript ID: materials-2271180). According to your and the reviewer’s professional suggestions, we have carefully made a revision addressing all the issues/comments in the list of responses. The changes in the revised version have been highlighted.
We hope the revised manuscript will meet your requirements and be accepted for publication.
Thank you very much for all your efforts in reviewing this manuscript.
Best wishes,
Yours sincerely,
Dr. Sicong Zhao
Reviewer 1:
Question: The microstructural characterization is excellent. However, I was unable to grasp the work's purpose. In the soultionized conditions, there are undissolved precipitates. Why did the authors select 545 degrees Celsius as the solutionizing temperature? Why was no aging treatment performed? Why has strength remained unchanged while ductility has improved?
Response: 1) Heat treatment is an important method to improve the mechanical properties of cast Mg-RE alloys, in which the effect of solid solution treatment directly determines the final properties of the alloy. Compared with the traditional Zr refined Mg alloys, Al refined Mg alloys have special microstructural characteristics.
Both undissolved phase and precipitated phase existed in matrix during the solid solution process of Al refined Mg alloys, which will significantly affect the solid solution effect, so it is necessary to study the microstructure evolution of Al refined alloy during solid solution treatment.
2) In order to avoid overheating, the solution treatment of Mg-RE alloys should be selected below the eutectic solution temperature. According to the DSC results, the eutectic solution temperature of Mg-3Y-4Nd-2Al alloy is 547.6 ℃. In addition, undissolved precipitates are Al2RE phase. These intermetallic compounds are stable phases formed after Al was added as the refiner, and they have high melting points (Al2Nd is 1460 ℃, Al2Y is 1490 ℃).
3) This paper focuses on the microstructure evolution in the solid solution process, which is the research basis of aging treatment. Aging treatment will be carried out in our follow-up study.
4) The reasons for the no significant change in strength after the process of solid solution treatment are mainly attributed to two aspects. On the one hand, the second strengthening effect caused by the dissolution of the second phase is weakened. On the other hand, the increase of solute elements in the matrix resulted in the enhancement of solid solution strengthening effect. The combined effect of the above positive and negative factors contributed to the no noticeable change in mechanical properties. The improvement of plasticity is related to the second phase in the alloy: first, there are large size and brittle intermetallic compounds at the grain boundary of the as-cast alloy. Stress concentration can easily occur in the eutectic phase of Mg-RE during tensile loading, which led to crack initiation and propagation, and the cast alloy shows the worst plasticity. With the extension of solution time, the eutectic phase gradually dissolved and the plasticity increased. After solution for 4 h, a large number of lamellar phases precipitated in the alloy, and the stress concentration was also easy to form at the sharp corner, which affected the plasticity of the alloy. With the further extension of the solution time, the lamellar precipitated phase decreased gradually, and the plasticity of the alloy increases gradually.
Reviewer 2 Report
The present research investigated the Effect of Solution Treatment Time on Microstructure Evolution and Properties of Mg-3Y-4Nd-2Al Alloy. The results are suitable to be considered for publication, however some kinds of improvement should be applied.
1) There are some English typos that needs to be corrected. For example:
- In the all text it is stated that "for different time", however "for different times" is true.
-Considering the part "All raw materials were smelted in resistance furnace with steel crucible", "in a resistance" is correct.
-Considering part "In general, the undissolved second phase at the grain boundary could hinders the growth of grains in the process of heat treatment", the "could hinder" is correct.
-In "With the extension of solution time, the yield strength and tensile strength change little, and the elongation increases greatly", "tensile strength changes a little" is true.
-At the end of Figure 10 caption, a "." is missed.
2) In the first paragraph of the "Introduction" it is mentioned that"Grain refinement is necessary for cast Mg-RE alloy.". However, it is not clarified why grain refinement is necessary?
3) Why 545 °C was used for solution treatment? Why other temperatures are not suitable?
4) What are the tensile samples dimension?
5) As an important weakness, there is no comparison between the present study results with other reports in the literature.
6) In fig. 2, there is no β1 phase, but it is stated inside the figure.
7) Regarding the paert "The morphology of granular Al2RE phase ("A"phase) and acicular Al11RE3 phase ("B" phase) was almost unchanged, while a large number of fine lamellar phase ("C" phase) was precipitated inside the grain boundary", what is this phase chemical composition?And why it was not observed in XRD pattern?
8) In terms of "The strengthening mechanism of as-cast alloy is mainly fine grain strengthening and second phase strengthening.", the grain size in as-cast sample is about 49 μm, so it is not true to state that fine grain strengthening is the hardening mechanism.
9) Considering the sentence "At the same time, the solid solution strengthening effect is enhanced due to the increase of solute elements in the matrix.", how the authors can confirm the increase in the solid solution elements in the matrix? It is necessary to analyze the chemical composition of the matrix by methods such as EDS.

Author Response
Dear Editor and reviewers:
Thanks a lot for your precious and constructive comments for improving the manuscript entitled "Effect of Solution Treatment Time on Microstructure Evolution and Properties of Mg-3Y-4Nd-2Al Alloy" (Manuscript ID: materials-2271180). According to your and the reviewer’s professional suggestions, we have carefully made a revision addressing all the issues/comments in the list of responses. The changes in the revised version have been highlighted.
We hope the revised manuscript will meet your requirements and be accepted for publication.
Thank you very much for all your efforts in reviewing this manuscript.
Best wishes,
Yours sincerely,
Dr. Sicong Zhao
Reviewer 2:
Question 1: There are some English typos that needs to be corrected. For example:
- In the all text it is stated that "for different time", however "for different times" is true.
-Considering the part "All raw materials were smelted in resistance furnace with steel crucible", "in a resistance" is correct.
-Considering part "In general, the undissolved second phase at the grain boundary could hinders the growth of grains in the process of heat treatment", the "could hinder" is correct.
-In "With the extension of solution time, the yield strength and tensile strength change little, and the elongation increases greatly", "tensile strength changes a little" is true.
-At the end of Figure 10 caption, a "." is missed.
Response: Thanks for your comments. We have corrected these grammatical errors in the text. The full manuscript was examined carefully, and the similar errors have been corrected.
Question 2: In the first paragraph of the "Introduction" it is mentioned that "Grain refinement is necessary for cast Mg-RE alloy." However, it is not clarified why grain refinement is necessary?
Response: In order to clarify this, we have added explanations and modified the manuscript accordingly. The cast Mg-RE alloys without refinement have very well developed and coarse grains, making their mechanical properties low while promoting the formation of defects such as hot cracking and porosity. Therefore, grain refinement is necessary for the cast Mg-RE alloy.
Question 3: Why 545 °C was used for solution treatment? Why other temperatures are not suitable?
Response: In order to avoid overheating, the solution treatment of Mg-RE alloy should be selected below the eutectic solution temperature. According to the DSC results, the eutectic solution temperature of Mg-3Y-4Nd-2Al alloy is 547.6 ℃. We added the DSC results in Section 3.2 of the revised manuscript, and explained the basis for selecting the solution temperature.
Question 4: What are the tensile samples dimension?
Response: The gauge dimension of the tensile specimens are 15mm×3 mm×2 mm. The tensile samples dimension has been added to the experimental method of the revised manuscript.
Question 5: As an important weakness, there is no comparison between the present study results with other reports in the literature.
Response: Thanks for your comment. We have added the comparison with the literature in the Section 3.3 Phase Evolution of Solution-treated Alloy in the revised manuscript.
Question 6: In fig. 2, there is no β1 phase, but it is stated inside the figure.
Response: Thank you for your reminding. The “β1” in Figure 2 (Figure 3 in the revised manuscript) has been deleted.
Question 7: Regarding the part "The morphology of granular Al2RE phase ("A"phase) and acicular Al11RE3 phase ("B" phase) was almost unchanged, while a large number of fine lamellar phase ("C" phase) was precipitated inside the grain boundary", what is this phase chemical composition? And why it was not observed in XRD pattern?
Response: The XRD and HAADF-SEM results showed that the fine lamellar phase precipitates were Al2RE phases containing Al, Nd and Y elements.
Question 8: In terms of "The strengthening mechanism of as-cast alloy is mainly fine grain strengthening and second phase strengthening.", the grain size in as-cast sample is about 49 μm, so it is not true to state that fine grain strengthening is the hardening
Response: When the grain size is 49 μm, calculated by Hall-Petch relation, the contribution value of fine grain strengthening of as-cast alloy is 40 MPa, and the contribution value of second phase strengthening is 69 MPa, so it is considered that fine grain strengthening is also the main strengthening mechanism.
Question 9: Considering the sentence "At the same time, the solid solution strengthening effect is enhanced due to the increase of solute elements in the matrix.", how the authors can confirm the increase in the solid solution elements in the matrix? It is necessary to analyze the chemical composition of the matrix by methods such as EDS.
Response: According to the EDS results, the total amount of RE elements in the as-cast alloy matrix is 0.25 at%, and the total amount of RE elements in the matrix after solution treatment is 0.62%, and thus the solid solute elements in the matrix are increased, which has been supplemented by the results of EDS in the Supporting Information.

Round 2
Reviewer 2 Report
-